# Advanced Strategy of Trophoblasts Retrieval and Isolation from the Cervix (TRIC): Comparison of Two HLA-G Antibodies for Immunomagnetic Isolation of Trophoblasts

**DOI:** 10.3390/jpm13010022

**Published:** 2022-12-22

**Authors:** Kirim Hong, Hee Yeon Jang, Sung Han Shim, Hee Young Cho, Dong Hyun Cha

**Affiliations:** 1Department of Obstetrics and Gynecology, CHA Gangnam Medical Center, CHA University, Seoul 06125, Republic of Korea; 2Department of Biomedical Science, College of Life Science, CHA University, Seongnam 13488, Republic of Korea

**Keywords:** trophoblasts retrieval and isolation from the cervix, extravillous trophoblast cells, HLA-G antibody, β-hCG, immunoisolation

## Abstract

Trophoblasts retrieval and isolation from the cervix (TRIC) is a non-invasive method which enables analysis of fetal genetic information from the extravillous trophoblast cells (EVTs). The aim of this study was to compare the efficacy of the HLA-G antibodies—G233 and 4H84—in isolating EVT cells and provide an optimized protocol of TRIC. We analyzed EVTs from 23 pregnant women in between 5 to 20 weeks of gestation who underwent invasive prenatal testing. Two HLA-G antibodies—G233 and 4H84—were used in a subgroup of 11 and 12 samples for immunomagnetic isolation. Cells with β-hCG expression were counted to compare the rate of isolated trophoblast cells. The rate of β-hCG positive cells was significantly different between the G233 and the 4H84 by immunefluorescence microscopy (*p* < 0.001). The percentage of β-hCG expressing cells in G233 and 4H84 groups were 62.4 ± 8.24% and 82.6 ± 7.1%, respectively (*p* < 0.001). The average fetal cell positive rate was 14.1 ± 3.78 in the G233 group while it was 25.8 ± 3.9 in the 4H84 group by fluorescence in situ hybridization (FISH) (*p* = 0.011). Immunoisolation of trophoblast cells using 4H84 HLA-G antibody was more efficient in capturing EVT cells than using G233 for successful clinical application of TRIC.

## 1. Introduction

Non-invasive prenatal diagnosis of aneuploidy has become one of the great interests in latest genetics field. Developing an efficient and reproducible method would offer clear advantages over invasive methods such as chorionic villous sampling (CVS) and amniocentesis, which are related to procedure-related miscarriage rate of 0.1–2% [1]. Up to date, the major source materials investigated for noninvasive prenatal screening are cell-free fetal DNA in the maternal blood and fetal cells obtained from the cervix. Non-invasive prenatal testing (NIPT) using cell-free DNA in maternal blood, can screen for the most common chromosome number disorders (13, 18, 21, X, Y) beginning around 10 weeks of pregnancy [2]. Although it can detect trisomies 13, 18, 21 with high detection rate of 98–99% with false positive rates of ~0.13% [3], it is not a diagnostic test due to several limiting factors. Mainly, maternal blood contains only a small fraction of fetal cells (4–10%) which originate from the placenta. In order to the test to be reliably sequenced and interpreted, fetal fraction in the study sample should exceed 4% of the total cell-free DNA in maternal plasma, which can be difficult to achieve [4]. Therefore, to overcome the limitations, another method investigated to non-invasively retrieve fetal cells is by obtaining trophoblast cells in maternal cervix. In early gestation, as placenta contacts the uterine deciduas, extravillous trophoblast (EVT) cells invade into uterine blood vessels and glands to open the luminal structures towards the intervillous space of the placenta. These EVT cells follow the endovascular, endoglandular, and interstitial invasion routes and are transported with glandular secretions into the uterine cavity and migrate towards the cervix [5]. Therefore, by using a cytobrush as in Pap smear, exfoliated trophoblast cells from the endometrial canal can be retrieved from the cervix safely and successfully [6,7] between 5 and 20 weeks of pregnancy [8]. These trophoblasts are isolated from maternal cells, using immunomagnetic nanoparticles, for analysis of fetal DNA, RNA, proteins and other useful genetic information [9]. Therefore, the immunomagnetic isolation of trophoblasts from maternal cells is one of the major key steps which determine accuracy of the method for its clinical application. During placenta development, cells express HLA-G as they differentiate from villous to extravillous, which makes HLA-G as a reliable phenotypic marker of differentiated EVT, [10] and it is also expressed by trophoblast cells captured in the endocervical canal [7]. Therefore, by using antibodies of HLA-G which is conjugated to magnetic nanoparticles (250 nm), EVT cells can be isolated from maternal origin cells. Out of several mouse monoclonal antibodies to HLA-G, G233 and 4H84 are known to show the highest immunohistochemical reaction in human placental tissue, [11] and therefore have been used in most of previous trophoblast retrieval studies. However, the difference between the two is that G233 binds to HLA-G1 isoform while 4H84 binds to HLA-G1 and G2 isoforms [12]. This can relate to the immunomagnetic capturing of EVT cells in TRIC since expression of HLA-G2 isoform is found to be limited to invasive trophoblastic phenotype (which characterizes EVT cells) while HLA-G1 is abundantly expressed in many other subpopulation of trophoblast cells. Therefore, it can be hypo-thesized that using 4H84 for immunomagnetic isolation in TRIC can capture higher percentage of EVT cells than using G233. Thus, the aim of this study was to compare the efficacy of the two widely used HLA-G antibodies—G233 and 4H84—in isolating EVT cells and provide an optimized protocol for successful clinical application of trophoblast retrieval and isolation from the cervix (TRIC).

## 2. Materials and Methods

### 2.1. Patient Selection

This study included a total of 23 patients who visited the Gangnam Cha Medical Center between 1 November 2018 and 31 April 2019. The inclusion criteria were normal, intrauterine, and singleton pregnancy of gestational age within 5–20 weeks. Patients with multiple pregnancies or active vaginal bleeding were excluded. We confirmed the fetal chromosome by CVS or amniocentesis. These invasive tests were performed due to clinical indications such as advanced maternal age with abnormal ultrasound finding, abnormal maternal serum analyze level for aneuploidy screening, or family history of aneuploidy. Pap test was performed before the invasive tests in all consenting pregnant women. These samples underwent TRIC using a basic four-step process (Figure 1). The protocol has been described in subsequent subsections. The Institutional Review Board of Gangnam Cha Medical Center (GCI-17-38) approved this study and all participating patients provided written informed consent.

### 2.2. Endocervical Sampling

The patient was put in the lithotomy position and a cytobrush was inserted through the external cervical os up to 2 cm into the cervical canal and was fully rotated for 360° to obtain sufficient cell mass. The samples were quickly immersed in PBS and were immediately transferred to the laboratory. To remove the mucus, samples were treated with 3% acetic acid (300 µL/10 mL) at room temperature for 5 min. After centrifugation at 900× *g* for 5 min at 4 °C, the cells were washed three times with cold PBS. Subsequently, the cells were fixed using 3.7% formalin for 10 min at 4 °C. Fixed cells were centrifuged at 900× *g* for 5 min, washed three times with cold PBS, counted, and stored immediately at 4 °C.

### 2.3. Immunomagnetic Isolation of Trophoblast Cells

Mouse anti-HLA-G antibodies, each of G233(10 µg/mL, Invitrogen, Waltham, MA, USA) and 4H84 (10 µg/mL, BD Biosciences, Pharminge, CA, USA), was incubated with 20 uL of 250 nm magnetic nanoparticles conjugated to a goat anti-mouse immunoglobulin G (IgG) antibody (Clemente Associates, Madison, CT, USA) overnight at 4 °C. Next day, the non-bound nanoparticles were washed three times with cold PBS in a magnetic strand. Then, the endocervical cells were resuspended in 1.5 mL PBS containing 1% Bovine serum albumin (BSA) and the anti-HLA-G antibody-coupled nanoparticles; they were incubated overnight again at 4 °C with mixing. The bound cells and non-bound cells were separated and collected in each tube using magnetic immobilization. The bound cells were washed three times with cold PBS in a magnetic strand. The bound cells which attached to the magnetic strand were considered as trophoblasts while the non-bound cells were considered as maternal cells.

### 2.4. Real Time PCR

Total RNA extraction from trophoblast cells isolated from cervical cells was extracted using RNeasy mini kit (QIAGEN, Hilden, Germany, 74106). Extracted using more than 1000 cells per sample, the method followed the manufacturer’s instructions. Total RNA was measured for purity and concentration using NanoDrop™ 2000 spectrophotometer (ND-2000, Thermo-Scientific, Waltham, MA, USA).

cDNA was synthesized from the extracted RNA, and SuperScript™ III First-Strand Synthesis System (18080051, Invitrogen) was used for cDNA synthesis. Synthesis of cDNA was according to the manufacturer’s instructions. Primers of the target gene were synthesized by Bioneer Co., Ltd. (Oakland, CA, USA). For real-time PCR, faststart SYBR green master kit (4673484001, Roche, Basel, Switzerland) was used, and CFX’s PCR Detection System (Bio-Rad, Hercules, CA, USA, 1855201) was used for analysis. After denaturing at 95 °C for 10 min, PCR conditions were repeated 39 times at 95°C for 15 s, at 60 °C for 1 min, and at 72 °C for 1 min. Gene expression analysis was normalized to non-binding of HLA-G type G233.

### 2.5. Immunofluorescence

For immunofluorescence microscopy, the isolated anti-HLA-G antibody-positive cells and anti-HLA-G antibody depleted cells were suspended in 200 µL of PBS on a slide and centrifuged at 1500 rpm for 5 min using the Cytospin 7620 (Wescor Inc., Logan, UT, USA). Cells attached to the slides were dried and blocked in PBS with 3% BSA at 4 °C for >1 h. The slides were incubated with β-hCG primary antibody (mouse, 10 μg/mL, 5H4-E2, Thermo Scientific, Waltham, MA, USA) or cytokeratin 7 (CK7) (rabbit, 10 ug/mL, ab181598, abcam, Cambridge, UK) overnight at 4 °C and washed three times using PBS containing 0.5% Tween 20 for 10 min at room temperature. Subsequently, the slide was incubated with Alexa Fluor^®^ 555 goat anti-mouse IgG (5 μg/mL, A-21422, Invitrogen Carlsbad, CA, USA) or Alexa Fluor^®^ 488 goat anti-rabbit IgG (1 mg, A-11034, Invitrogen Carlsbad, CA, USA) at 4 °C for 1 h and washed three times with PBS. The cells were then stained with 4′,6-diamidino-2-phenylindole dihydrochloride (1 μg/mL) at room temperature for 10 min and washed three times with PBS. The cells were mounted on slides with coverslip and observed under the Axio Imager 2 fluorescence microscope (Carl Zeiss, Thornwood, NY, USA). The percentage of cells expressing β-hCG was calculated.

### 2.6. Fluorescence In Situ Hybridization (FISH)

Isolated cells were incubated with fluorescence in situ hybridization (FISH) probes against chromosome X and Y: DXZ1 Alpha Satellite SpectrumOrange and DYZ1 satellite III SpectrumGreen were the X and Y chromosome probes (Abbott Molecular, Des Plaines, IL, USA), respectively. FISH was performed according to the manual and signals were analyzed using the Olympus BX85 microscope (Olympus, Shinjuku, Tokyo, Japan) with the CytoVision^®^ (Leica, Wetzlar, Germany) FISH imaging software.

### 2.7. Statistical Analysis

All statistical analyses were performed using R version 4.0.2. Data are expressed as mean ± standard deviation and the G233 group was compared with the 4H84 group using the student’s *t*-Test. *p*-values < 0.05 were considered statistically significant.

## 3. Results

This study compared the efficacy of the two widely used HLA-G antibodies in isolating EVT cells. The mean age of the subjects was 33.5 ± 4.1 and 34.3 ± 4.8 years in the G233 and the 4H84 groups, respectively (*p* = 0.673) (Table 1). Additionally, there was no statistical difference in gestational age at which samples were taken between the two groups (*p* = 0.943).

Expression of β-hCG and HLA-G in trophoblast cells with HLA-G antibody G233 and 4H84 were compared (Figure 2). We have obtained the results that the β-hCG and HLA-G, the specific gene expressed in trophoblast, is increased in binding cells and it is obtained by quantifying the results that trophoblast was sorted. Comparing the expression of β-hCG and HLA-G in the G233 and the 4H84 bound cells, 4H84 showed a higher binding rate than G233 with β-hCG (*p* = 0.501) and HLA-G (*p* = 0.049). 

We compared the expression level of β-hCG between G233 and 4H84 clones by immunofluorescence microscopy (Figure 3). The G233 and the 4H84 preparation revealed cytotrophoblast cells that were positively labeled by β-hCG (Figure 3A, arrows) and other cervical cells that were non-reactive to β-hCG (Figure 3A, asterisks). The rate of β-hCG positive cells was significantly different between the G233 and the 4H84. (*p* < 0.001) (Figure 3B).

The total number of endocervical cells obtained from the 23 samples ranged between 1.36 × 10^6^ and 1.49 × 10^6^ and was independent of the gestational age (*p* = 0.794). The number of fixed cells were 2.16 × 10^5^ and 2.39 × 10^5^ in the G233 and the 4H84 groups, respectively, with no statistically significant difference (*p* = 0.290). The number of HLA-G positive cells were 4299.3 and 3755.9 in the G233 and the 4H84 groups, respectively, which also showed no statistically significant difference (*p* = 0.838). The rate of β-hCG positive cells was significantly different between the two groups, as expected; the average rate was 62.4 ± 8.24% in the G233 group while it was 82.6 ± 7.1% in the 4H84 (*p* < 0.001), which proved the superiority of 4H84 HLA-G antibody over G233 in capturing trophoblast cells (Table 2).

Moreover, FISH method was carried out to reveal the presence of Y chromosome in isolated trophoblasts in the 7 samples that were identified as male fetuses by invasive tests, such as CVS or amniocentesis, to confirm the purity of the isolated trophoblast cells (Figure 4). Fetal cells can be distinguished from maternal cells using FISH probes for X (spectrum orange) and Y chromosome (spectrum green) in cells captured by antibodies (Figure 4A). A larger number of fetal cells were identified in the cells captured using the 4H84 antibody compared to G233 (Figure 4B). The average FISH positive rate was 14.1 ± 3.78 in the G233 group while it was 25.8 ± 3.9 in the 4H84 group (*p* = 0.011), which validated the higher efficacy of 4H84 HLA-G antibody compared to G233 in isolating trophoblasts (Table 3).

We compared β-hCG and CK7 expression pattern between G233 (top) and 4H84 (bottom) clones by immunofluorescence (Figure 5). More β-hCG and CK7 expressing extravillous trophoblast cells (arrow) were identified in the cells captured by the 4H84 antibody.

Comparison of expression patterns of HLA-G in trophoblast cells captured by G233 (top) and 4H84 antibody (bottom). More β-hCG and CK7 expressing cells were identified in the cells captured by the 4H84 antibody.

## 4. Discussion

The non-invasive approaches of prenatal diagnosis have been extensively investigated in recent decades which includes the isolation of fetal cell free DNA from maternal serum or the retrieval of trophoblasts from the maternal cervix. When these two methods are compared, TRIC exhibits some considerable advantages. Firstly, while NIPT is available from 8 weeks of gestation, TRIC is available earlier as 5 weeks of gestation. Moreover, NIPT can only extract fragmented fetal DNA while TRIC can retrieve intact fetal cell which contains the whole fetal DNA [5]. Furthermore, this study result showed that the average rate of β-hCG (trophoblast specific marker) positive cells from the transcervical samples ranged from 51.8% to 96.0%, regardless of the HLA-G Ab type used, which implicates significantly high percentage of fetal cells in these transcervical samples when compared to the average fetal fraction in maternal plasma, which is reported to be about 10% to 15% [13]. Owing to these valuable potentials of TRIC to be developed as another non-invasive prenatal screening method, studies regarding strategies for improving the rate of purity in isolation of fetal cells in the transcervical samples have been extensively researched over past years. Recent study in our institution has published an optimized fixation protocol in TRIC, which found out that immediate immersion of the samples in phosphate-buffered saline (PBS) and removing the mucus by 3% acetic acid treatment before fixation with formalin was most effective in achieving the highest purity rate of trophoblast isolation [14].

Therefore, as a continuous work to optimize each step in TRIC, this study was designed to focus on the immunomagnetic isolation technique in capturing trophoblasts. HLA-G is a reliable phenotypic marker of differentiated EVT since it is expressed in trophoblast cells as they differentiate into extravillous from villi; in other words, as the trophoblasts develop to invade into decidua basalis within the stroma, vessels, and uterine glands, HLA-G expression becomes positive [10]. Therefore, the trophoblast cells collected at the endocervical canal also express HLA-G, which allows them to be distinguished from maternal origin cells that are collected together during the sampling of TRIC. To be more specific, the immunomagnetic isolation of trophoblasts uses mouse anti-HLA-G antibodies that are coupled with magnetic nanoparticles conjugated to goat anti-mouse IgG antibodies (Figure 1). Since only trophoblast express HLA-G, capturing those cells that are bound to the HLA-G antibodies allows sorting out the pure trophoblasts from maternal cells in the sample.

There are several types of HLA-G antibodies that are known to show positive immunofluorescene staining in extravillous trophoblasts, which include G233 and 4H84 [11]. Therefore, this study intended to compare the efficacy in trophoblast isolation between the two HLA-G antibodies. HLA-G has seven isoforms (HLA-G1 to G7), which consists of different combinations of α1, α2, and α3 domains of heavychain and β2-microglobulin in extracellular regions [15]. There is a major difference between the two HLA-G antibodies. G233 binds to native HLA-G1 isoform but not to denatured HLA-G1 molecules or both the native and denatured forms of HLA-G2. On the other hand, 4H84 binds to native HLA-G2, as well as the denatured isoform of HLA-G1 and HAL-G2 [12]. This can significantly affect the efficiency of immunomagnetic capturing of EVT cells in TRIC since expression of HLA-G2 isoform is found to be limited to invasive trophoblastic phenotype (which characterizes EVT cells) while HLA-G1 is abundantly expressed in many other subpopulation of trophoblast cells. Therefore, as it was hypothesized, this study result proved that using 4H84 for immunomagnetic isolation in TRIC can capture higher percentage of EVT cells than using G233.

Cytokeratin 7 (CK7), the specific epithelial intermediate filament, also is expressed in the isolated cytotrophoblast cells and the best markers for distinguishing of isolated trophoblast [16]. The cells captured by the 4H84 antibody expressed more β-hCG and CK7 expressing cells than the cells with G233 antibody in our study. Thess results support that 4H84 antibody is more effective than G233 antibody in isolating EVT cells.

Although a limitation of this study exists due to its small sample size, it has some considerable strength since it is the first novel study to compare the conventionally used HLA-G antibodies in immunoisolation of trophoblasts. This study contributes to the establishment of optimized protocol of TRIC which has promising future of clinical application.

## 5. Conclusions

Immunoisolation of trophoblast cells using 4H84 HLA-G antibody was more efficient in capturing EVTs than using G233. This result provides a valuable update on the optimized protocol for successful clinical application of TRIC. Future studies on strategies in eliminating the HLA-G Ab captured cells that are non-trophoblasts will be required to further improve the purity rate of trophoblast isolation.

## Figures and Tables

**Figure 1 jpm-13-00022-f001:**
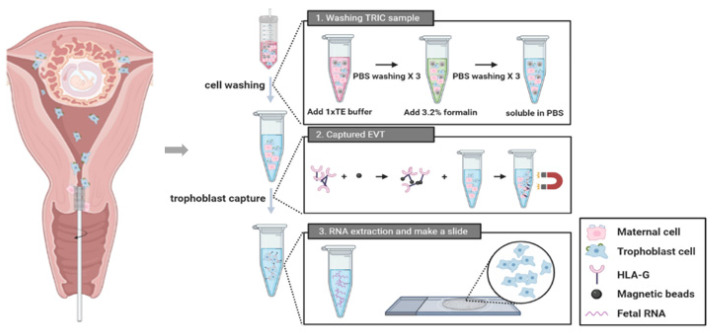
A procedure of trophoblast retrieval and isolation from the cervix (TRIC).

**Figure 2 jpm-13-00022-f002:**
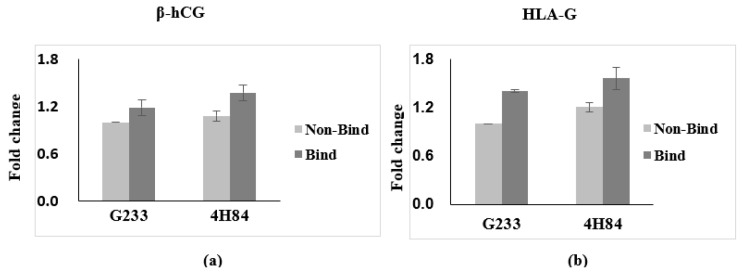
Expression of β-hCG and HLA-G in trophoblast cells with HLA-G antibody capture. (**a**) Expression of β-hCG in trophoblast cells increased with more 4H84 antibody than G233 antibody. (**b**) Expression of HLA-G in trophoblast cells increased with more 4H84 antibody than G233 antibody.

**Figure 3 jpm-13-00022-f003:**
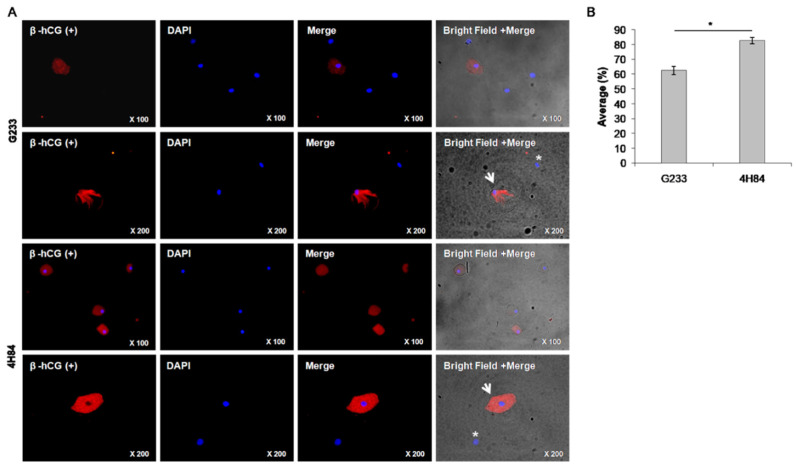
β-hCG expression between G233 and 4H84 clones by Immunofluorescence. (**A**) Comparison of expression patterns of HLA-G in trophoblast cells captured by G233 (top two rows) and 4H84 antibody (bottom two rows). More β-hCG expressing cells were identified in the cells captured by the 4H84 antibody. (**B**) β-hCG expression type of G233 and 4H84 clone by cell count percentage. (Arrow: extravillous trophoblast. Asterisks: immune cell).

**Figure 4 jpm-13-00022-f004:**
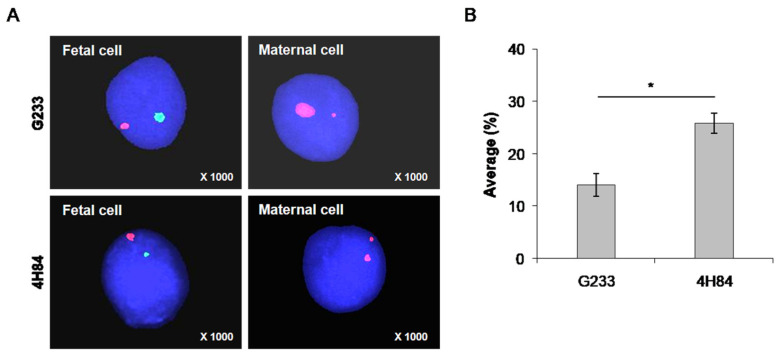
Distinguishing fetal cells from maternal cells using FISH probes for X (spectrum orange) and Y chromosome (spectrum green) in cells captured by antibodies. (**A**) FISH analysis results of cells captured using G233 and 4H84 antibodies. A larger number of fetal cells were identified in the cells captured using the 4H84 antibody. (**B**) Percentage of FISH signals-XY probe type of G233 and 4H84 clone by cell count. (*: *p*-value < 0.05).

**Figure 5 jpm-13-00022-f005:**
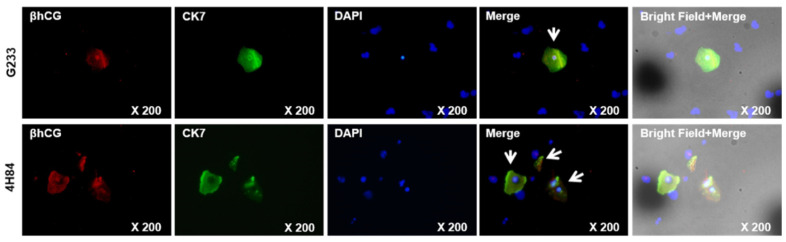
β-hCG and CK7 expression between G233 and 4H84 clones by Immunofluorescence.

**Table 1 jpm-13-00022-t001:** Maternal basic characteristics according to HLA-G Ab type.

HLA-G Type	G233 (*n* = 11)	4H84 (*n* = 12)	*p*-Value
Maternal Age (yrs)	33.5 ± 4.1	34.3 ± 4.8	0.673
GA (days)	69.8 ± 23.79	69.3 ± 24.1	0.943
Gravidity	1.9 ± 1.04	2 ± 0.95	0.830
Parity	0.6 ± 0.52	0.6 ± 0.51	0.558
BMI (kg/m^2^)	20.3 ± 2.49	21.4 ± 3.07	0.347

GA; gestational age, BMI; body mass index.

**Table 2 jpm-13-00022-t002:** Trophoblast contents and detection of β -hCG.

HLA Type	G233 (*n* = 11)	4H84 (*n* = 12)	*p*-Value
Endocervical cell	1.36 × 10^6^ ± 1.1 × 10^6^	1.49 × 10^6^ ± 1.2 × 10^6^	0.794
Fixed cell	2.16 × 10^5^ ± 4.3 × 10^4^	2.39 × 10^5^ ± 5.7 × 10^4^	0.290
HLA-G positive cell	4299.3 ± 5440	3755.9 ± 6958	0.838
β-hCG positive rate (%)	62.4 ± 8.24	82.6 ± 7.1	<0.001

**Table 3 jpm-13-00022-t003:** Trophoblast contents using fluorescence in situ hybridization.

HLA Type	G233 (*n* = 3)	4H84 (*n* = 4)	*p*-Value
GA(days)	62.3 ± 15.7	75 ± 14.5	0.333
Fetal sex	M	M	-
Karyotype (genetic test)	XY	XY	-
Fetal Sex (after delivery)	XY	XY	-
FISH(X/Y) rate (%)	14.1 ± 3.78	25.8 ± 3.9	0.011

GA; gestational age, FISH; fluorescence in situ hybridization.

## Data Availability

Not applicable.

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
