# Peer review of "Advanced Strategy of Trophoblasts Retrieval and Isolation from the Cervix (TRIC): Comparison of Two HLA-G Antibodies for Immunomagnetic Isolation of Trophoblasts"

_jpm, 2022, doi:10.3390/jpm13010022_

Round 1
Reviewer 1 Report
This study aimed to compare the efficacy in trophoblast isolation between the two HLA-G antibodies and found that 4H84 antibody is more effective than G233 265 antibody in isolating EVT cells. This result would be helpful for clinnical and laboratory options,but there are some problems that require the authors' attention:
1.The innovation of the article is not strong, and the workload is not enough to support the publication of an article. Suggest the author detect several more HLA antibodies.
2.As the authors stated, the sample size of the study was too small, we thought it could be increased.
3.The table of the article could actually be compressed and expressed in a more concise form, but maybe because the sample size was too small, the author divided it into several tables.
4.Lines 166 (P<0.049) and 184 (2.16X105 and 2.39X105) of the article are formatted incorrectly, note that they should be consistent with the previous article.
Author Response
Comments and Suggestions for Authors
This study aimed to compare the efficacy in trophoblast isolation between the two HLA-G antibodies and found that 4H84 antibody is more effective than G233 265 antibody in isolating EVT cells. This result would be helpful for clinnical and laboratory options,but there are some problems that require the authors' attention:
1.The innovation of the article is not strong, and the workload is not enough to support the publication of an article. Suggest the author detect several more HLA antibodies.
ïƒ Thank you for your comment!
G233 and 4H84 were used in this study as the most frequently used HLA-G antibodies as a result of literature review. As a follow-up research, we are planning a study by increasing various antibodies and sample sizes.
2.As the authors stated, the sample size of the study was too small, we thought it could be increased.
-> Thank you for your suggestion!
We need time to collect enough samples, so we are planning a study with more samples and more types of HLA-G antibodies.
3.The table of the article could actually be compressed and expressed in a more concise form, but maybe because the sample size was too small, the author divided it into several tables.
-> Thank you for your comment!
We compressed the table and showed in a more concise form as below.
Table 1. Maternal basic characteristics according to HLA-G Ab type
HLA-G type |
G233 (n=11) |
4H84 (n=12) |
p-value |
Maternal Age (yrs) |
33.5 ± 4.1 |
34.3 ± 4.8 |
0.673 |
GA (days) |
69.8 ± 23.79 |
69.3 ± 24.1 |
0.943 |
Gravidity |
1.9 ± 1.04 |
2 ± 0.95 |
0.830 |
Parity |
0.6 ± 0.52 |
0.6 ± 0.51 |
0.558 |
BMI (kg/m2) |
20.3 ± 2.49 |
21.4 ± 3.07 |
0.347 |
Table 2. Trophoblast contents and detection of β -hCG.
HLA type |
G233 (n=11) |
4H84 (n=12) |
p-value |
Endocervical cell |
1.36X106 ± 1.1X106 |
1.49X106 ± 1.2X106 |
0.794 |
Fixed cell |
2.16X105 ± 4.3X104 |
2.39X105 ± 5.7X104 |
0.290 |
HLA-G positive cell |
4299.3 ± 5440 |
3755.9 ± 6958 |
0.838 |
β -hCG positive rate (%) |
62.4 ± 8.24 |
82.6 ± 7.1 |
<0.001 |
Table 3. Trophoblast contents using fluorescence in situ hybridization
HLA type |
G233 (n=3) |
4H84 (n=4) |
p-value |
GA(days) |
62.3 ± 15.7 |
75 ± 14.5 |
0.333 |
Fetal sex |
M |
M |
- |
Karyotype (genetic test) |
XY |
XY |
- |
Fetal Sex (after delivery) |
XY |
XY |
- |
FISH(X/Y) rate (%) |
14.1 ± 3.78 |
25.8 ± 3.9 |
0.011 |
4.Lines 166 (P<0.049) and 184 (2.16X105 and 2.39X105) of the article are formatted incorrectly, note that they should be consistent with the previous article.
-> Thank you for your comment!
We corrected these as below
(P<0.049) -> (P=0.049) and (2.16X105 and 2.39X105) -> (2.16X105 and 2.39X105)

Reviewer 2 Report
Interesting topic, sound presentation and justification.
My only concern is related to the sampling method applied.
1. Could you please elaborate how the sample was selected?
2. If the sample size would affecr the validity of the results?
3. Finally, allow me a suggetion. You could possibly apply (M)ANOVA instead of t-test?
Author Response
Comments and Suggestions for Authors
Interesting topic, sound presentation and justification.
My only concern is related to the sampling method applied.
- Could you please elaborate how the sample was selected?
ïƒ Thank you for your question.
Since the amount of each sample was insufficient to test the two antibodies, the collected samples were randomly assigned and the experiment was conducted.
- If the sample size would affecr the validity of the results?
-> Thank you for your comment!
The test results of each sample consistently showed that more trophoblast cells were collected with 4H84 antibody. Therefore, we do not think that the sample size affected the validity of the test results. However, we are preparing for a larger scale study with more samples and more diverse HLA-G antibodies.
- Finally, allow me a suggetion. You could possibly apply (M)ANOVA instead of t-test?
-> Thank you for your suggestion!
As a result of consulting with medical statistician, we did a two-way ANOVA. The new results are attached below. There was no difference between the previous results and new one.
(Prof. Inkyung Jung, Division of Biostatistics, Department of Biomedical Systems Informatics, Yonsei University College of Medicine, Seoul, South Korea)
Table 1. Maternal basic characteristics according to HLA-G Ab type
HLA-G type |
G233 ( n=11) |
4H84 (n=12) |
p-value |
Maternal Age (yrs) |
33.5 ± 4.1 |
34.3 ± 4.8 |
0.907 |
GA (days) |
69.8 ± 23.79 |
69.3 ± 24.1 |
0.987 |
Gravidity |
1.9 ± 1.04 |
2 ± 0.95 |
0.724 |
Parity |
0.6 ± 0.52 |
0.6 ± 0.51 |
0.558 |
BMI (kg/m2) |
20.3 ± 2.49 |
21.4 ± 3.07 |
0.450 |
Table 2. Trophoblast contents and detection of β -hCG.
HLA type |
G233 ( n=11) |
4H84 (n=12) |
p-value |
Endocervical cell |
1.36X106 ± 1.1X106 |
1.49X106 ± 1.2X106 |
0.726 |
Fixed cell |
2.16X105± 4.3X104 |
2.39X105±5.7 X104 |
0.335 |
HLA-G positive cell |
4299.3±5440 |
3755.9±6958 |
0.154 |
β -hCG Positive rate (%) |
62.4 ± 8.24 |
82.6 ± 7.1 |
<0.001 |
Table 3. Trophoblast contents using fluorescence in situ hybridization
HLA type |
G233 ( n=3) |
4H84 (n=4) |
p-value |
GA |
62.3 ± 15.7 |
75 ± 14.5 |
0.185 |
Fetal sex |
M |
M |
- |
Karyotype (genetic test) |
XY |
XY |
- |
Fetal Sex (after delivery) |
XY |
XY |
- |
FISH(X/Y) rate (%) |
14.1 ± 3.78 |
25.8 ± 3.9 |
0.039 |

Reviewer 3 Report
This is a mostly well written manuscript. Non-invasive prenatal diagnosis based on cervical trophoblast cells has a wide application prospect. There is still a lack of simple, standard, stable, safe and reliable methods for routine clinical analysis. It is difficult to apply the technique to clinical laboratory diagnosis on a large scale. We need more researches on this. The development of minimally invasive prenatal diagnostic methods for clinical applications is the inevitable trend of future development. The isolation of trophoblast cells is an important step of minimally invasive prenatal diagnosis based on cervical exfoliated trophoblast cells. This study provides a valuable method for non-invasive prenatal diagnosis. It is suggested that the pregnancy outcome of pregnant women should be provided to reduce the concern of using this technology.
1. In the current research methods, trophoblast cells were collected from the cervical canal of pregnant women by different ways from 5-7 to 13-15 weeks. It is generally believed that more fetal trophoblasts can be obtained in the early stage of pregnany. This study collected trophoblast cells from 5-20 weeks pregnant women, with a maximum of 16 + 2 weeks in the list. Why did you choose these weeks? What is the basis?
2.The sample size is small, we suggest to expand the sample capacity.
3.Fetal identification of isolated cells is an essential step in prenatal diagnosis using trophoblast cells. The isolated cells were identified as fetal cells by morphological, immunophenotypic and genetic information analysis. But,this study lacks morphological reserch methods. It is useful to know which of the following Cytotrophoblastic cells, intercytotrophoblast, syncytiotrophoblast cells were isolated by 4H84 for further study.
4.The methods are well described and the results are clear. The results of this study indicate that the isolation of trophoblast by immunization with 4H84 HLA-G antibody is more effective in capturing EVTs than with G233 antibody. English language and style are good, just need a slight spell check.
Author Response
Comments and Suggestions for Authors
This is a mostly well written manuscript. Non-invasive prenatal diagnosis based on cervical trophoblast cells has a wide application prospect. There is still a lack of simple, standard, stable, safe and reliable methods for routine clinical analysis. It is difficult to apply the technique to clinical laboratory diagnosis on a large scale. We need more researches on this. The development of minimally invasive prenatal diagnostic methods for clinical applications is the inevitable trend of future development. The isolation of trophoblast cells is an important step of minimally invasive prenatal diagnosis based on cervical exfoliated trophoblast cells. This study provides a valuable method for non-invasive prenatal diagnosis. It is suggested that the pregnancy outcome of pregnant women should be provided to reduce the concern of using this technology.
- In the current research methods, trophoblast cells were collected from the cervical canal of pregnant women by different ways from 5-7 to 13-15 weeks. It is generally believed that more fetal trophoblasts can be obtained in the early stage of pregnany. This study collected trophoblast cells from 5-20 weeks pregnant women, with a maximum of 16 + 2 weeks in the list. Why did you choose these weeks? What is the basis?
ïƒ Thank you for your comment!
As you mentioned, more fetal trophoblasts can be obtained in the early pregnancy.
We decided to collect samples at various weeks from 5 to 20 weeks to see how the number of trophoblast cells obtained differs according to gestational age.
2.The sample size is small, we suggest to expand the sample capacity.
ïƒ Thank you for your suggestion!
We need time to collect enough samples, so we are planning a study with more samples and more types of HLA-G antibodies.
3.Fetal identification of isolated cells is an essential step in prenatal diagnosis using trophoblast cells. The isolated cells were identified as fetal cells by morphological, immunophenotypic and genetic information analysis. But,this study lacks morphological reserch methods. It is useful to know which of the following Cytotrophoblastic cells, intercytotrophoblast, syncytiotrophoblast cells were isolated by 4H84 for further study.
ïƒ Thank you for your suggestion!
We are planning to investigate morphological analysis of fetal cells for further study.
4.The methods are well described and the results are clear. The results of this study indicate that the isolation of trophoblast by immunization with 4H84 HLA-G antibody is more effective in capturing EVTs than with G233 antibody. English language and style are good, just need a slight spell check.
ïƒ Thank you for your comment!
We checked the spell in the manuscript thoroughly.
